# The role of history and strength of the oceanic forcing in sea-level projections from Antarctica with the Parallel Ice Sheet Model

Ronja Reese[1], Anders Levermann[1,2,3], Torsten Albrecht[1], Hélène Seroussi[4], and Ricarda Winkelmann[1,2]

[1] Potsdam Institute for Climate Impact Research (PIK), Member of the Leibniz Association, P.O. Box 60 12 03, 14412 Potsdam, Germany

[2] University of Potsdam, Institute of Physics and Astronomy, Karl-Liebknecht-Str. 24-25, 14476 Potsdam, Germany

[3] LDEO, Columbia University, New York, USA

[4] Jet Propulsion Laboratory, California Institute of Technology, Pasadena, CA, USA

**Correspondence:** Ronja Reese (ronja.reese@pik-potsdam.de)

**Abstract.** Mass loss from the Antarctic Ice Sheet constitutes the largest uncertainty in projections of future sea-level rise. Ocean-driven melting underneath the floating ice shelves and subsequent acceleration of the inland ice streams is the major reason for currently observed mass loss from Antarctica and is expected to become more important in the future. Here we show that for projections of future mass loss from the Antarctic Ice Sheet, it is essential (1) to better constrain the sensitivity of sub-shelf melt rates to ocean warming, and (2) to include the historic trajectory of the ice sheet. In particular, we find that while the ice-sheet response in simulations using the Parallel Ice Sheet Model is comparable to the median response of models in three Antarctic Ice Sheet Intercomparison projects – initMIP, LARMIP-2 and ISMIP6 – conducted with a range of ice-sheet models, the projected 21st century sea-level contribution differs significantly depending on these two factors. For the highest emission scenario RCP8.5, this leads to projected ice loss ranging from $1.4$ to $4.0\,\mathrm{cm}$ of sea-level equivalent in simulations in which ISMIP6 ocean forcing drives the PICO ocean box model where parameter tuning leads to a comparably low sub-shelf melt sensitivity and in which no surface forcing is applied. This is opposed to a likely range of $9.1$ to $35.8\,\mathrm{cm}$ using the exact same initial setup, but emulated from the LARMIP-2 experiments with a higher melt sensitivity, even though both projects use forcing from climate models and melt rates are calibrated with previous oceanographic studies. Furthermore, using two initial states, one with a previous historic simulation from 1850 to 2014 and one starting from a steady state, we show that while differences between the ice-sheet configurations in 2015 seem marginal at first sight, the historic simulation increases the susceptibility of the ice sheet to ocean warming, thereby increasing mass loss from 2015 to 2100 by 5 to 50%. Hindcasting past ice-sheet changes with numerical models would thus provide valuable tools to better constrain projections. Our results emphasize that the uncertainty that arises from the forcing is of the same order of magnitude as the ice-dynamic response for future sea-level projections.

# 1 Introduction

Observations show that the Antarctic Ice Sheet is currently not in equilibrium and that its contribution to global sea-level rise is increasing (Shepherd et al., 2018). Its future contribution is the largest uncertainty in sea-level projections (Pörtner et al., 2019) with its evolution driven by snowfall increases (e.g., Ligtenberg et al., 2013; Frieler et al., 2015) that are counteracted by increased ocean forcing (e.g., Hellmer et al., 2012; Naughten et al., 2018) and potentially instabilities such as the Marine Ice Sheet Instability (Weertman, 1974; Schoof, 2007) and the Marine Ice Cliff Instability (DeConto and Pollard, 2016).

In recent years, sea-level projections of the Antarctic Ice Sheet were conducted with individual ice-sheet models (e.g., DeConto and Pollard, 2016; Golledge et al., 2019) and extended by comprehensive community efforts such as the Ice Sheet Model Inter-comparison Project for CMIP6 (ISMIP6, Nowicki et al., 2016, in prep; Seroussi et al., under review) and the Linear Antarctic Response Model Intercomparison Project (LARMIP-2, Levermann et al., 2014, 2020) projects. In ISMIP6, a protocol for Antarctic projections was developed and ice-sheet model responses to oceanic and atmospheric forcing from selected CMIP5 models (Barthel et al., 2019) were gathered and compared for the first time. As a first step of ISMIP6, initMIP-Antarctica did test the effect of different model initialisation on idealized experiments (Seroussi et al., 2019). While the response of the ice sheet to surface mass balance forcing was similar among the models, they showed very different responses to basal melt rate changes. Similarly, in ISMIP6 a large spread in model projections is found, with ice volume changes from $-7.8$ to $30.0\,\mathrm{cm}$ of Sea-Level Equivalent (SLE) under the highest greenhouse gas emission scenario (Representative Concentration Pathway RCP8.5) with the largest uncertainties coming from ocean-induced melt rates, the calibration of melt rates and the ice dynamic response to oceanic changes. The ISMIP6 projections are given with respect to control simulation, hence not considering current trends of mass loss.

Sea-level estimates in ISMIP6 are in many cases substantially lower than the ocean-driven mass loss projected by LARMIP-2. In LARMIP-2, the sea-level contribution of the Antarctic Ice Sheet is emulated from step-forcing experiments using linear response function theory (Winkelmann and Levermann, 2013). A median mass loss of $17\,\mathrm{cm}$ with a likely range from $9\,\mathrm{cm}$ to $36\,\mathrm{cm}$ and a very likely range of $6\,\mathrm{cm}$ to $58\,\mathrm{cm}$ is found. In contrast to ISMIP6, atmospheric changes, that add between $-2.5$ and $84.5\,\mathrm{mm}$ SLE depending on the CMIP5 forcing, are not considered in LARMIP-2, and we here also focus on the dynamic, ocean-driven response of the ice sheet.

In projections of the future Antarctic sea-level contribution following the ISMIP6 and LARMIP-2 protocols, oceanic forcing is obtained from sub-surface ocean conditions in general circulation models, e.g., from results of the fifth phase of the Coupled Model Intercomparison Project (CMIP5, Taylor et al., 2012). This approach takes into account that sub-shelf melt rates are mainly driven by inflow of ocean water masses at depth (Jacobs et al., 1992). However, CMIP5 models do not include ice-shelf cavities and related feedbacks that might increase the future oceanic forcing on the ice shelves (Timmermann and Goeller, 2017; Donat-Magnin et al., 2017; Bronselaer et al., 2018; Golledge et al., 2019). Ocean temperatures from CMIP5 models therefore have to be extrapolated into ice-shelf cavities (Jourdain et al., under review). Alternatively, output from high-resolution models that resolve ocean dynamics on the continental shelf and within the ice-shelf cavities could be used (e.g., Hellmer et al., 2012; Naughten et al., 2018).

The sub-surface ocean forcing informs parameterizations that provide melt rates underneath the ice shelves for ice-sheet models. For the ISMIP6 experiments, a depth-dependent, non-local parameterization and a depth-dependent, local parameterization have been proposed (Jourdain et al., under review) that both mimic a quadratic dependency of melt rates on thermal forcing (Holland et al., 2008). As an alternative, more complex modules that capture the basic physical processes within ice-shelf cavities have been developed recently (Lazeroms et al., 2018; Reese et al., 2018a). We here analyse results as submitted to ISMIP6 that apply the Potsdam Ice-shelf Cavity mOdel (PICO; Reese et al., 2018a) which extends the ocean box model (Olbers and Hellmer, 2010) for application in three-dimensional ice-sheet models. The model has been tested and compared to other parameterizations for an idealized geometry (Favier et al., 2019). In this case, the induced ice-sheet response matches the response driven by a three-dimensional ocean model. In contrast to ISMIP6, the LARMIP-2 experiments are forced by basal-melt rate changes directly. Scaling factors between global mean temperature changes and Antarctic sub-surface temperature changes are determined from CMIP5 models. These are used to generate ocean temperature forcing under different RCP scenarios emulated from MAGICC-6.0 RCP realisations (Meinshausen et al., 2011). Sub-shelf melt rates are assumed to increase by 7 to $16\,\mathrm{m\,a^{-1}}$ per degree of sub-surface ocean warming, based on Jenkins (1991) and Payne et al. (2007).

Here we compare simulations with the Parallel Ice Sheet Model as submitted to ISMIP6 with results obtained following the LARMIP-2 protocol and analyse (1) the effect of the oceanic forcing and (2) the effect of a historic simulation preceding the projections. In Sect. 2 we describe the methods used and the initial configurations of PISM. This is followed by an analysis of the experiments for ISMIP6 with only ocean forcing applied and the results obtained when following the LARMIP-2 protocol in Sect. 3. These are compared and discussed in Sect. 4 and 5.

## 2 Methods

We use the comprehensive, thermo-mechanically coupled Parallel Ice Sheet Model (PISM, Bueler and Brown, 2009; Winkelmann et al., 2011; the PISM authors, 2019) which employs a superposition of the Shallow-Ice and Shallow-Shelf Approximations (Hutter, 1983; Morland, 1987; MacAyeal, 1989). We apply a power-law relationship between SSA basal sliding velocities and basal shear stress with a Mohr–Coulomb criterion relating the yield stress to parameterized till material properties and the effective pressure of the overlaying ice on the saturated till (Bueler and Pelt, 2015). Basal friction and sub-shelf melting are linearly interpolated on a sub-grid scale around the grounding line (Feldmann et al., 2014). In order to improve the approximation of driving stress across the grounding line, the surface gradient is calculated using centered differences of the ice thickness across the grounding line. We apply eigen-calving (Levermann et al., 2012) in combination with the removal of ice that is thinner than $50\,\mathrm{m}$ or extends beyond present-day ice fronts (Fretwell et al., 2013).

### 2.1 Initial configurations

We use two model configurations of the Antarctic Ice Sheet that were submitted to ISMIP6, one with a preceding historic simulation from 1850 to 2014 and one starting from a steady-state. Both configurations share the same initialisation procedure: starting from Bedmap2 ice thickness and topography (Fretwell et al., 2013), a spin-up is run for $400,000$ years with constant

geometry to obtain a thermodynamic equilibrium with present-day climate on $16\,\mathrm{km}$. Based on this, an ensemble of simulations with varying model parameters is run for several thousand years towards dynamic equilibrium on $8\,\mathrm{km}$ horizontal resolution. The simulations employ 121 vertical layers with a quadratic spacing from $13\,\mathrm{m}$ at the ice shelf base to $100\,\mathrm{m}$ towards the surface. We vary parameters of PICO (heat exchange coefficient $\gamma_T$ and overturning coefficient $C$) as well as the minimum till friction angle in the parameterized till material properties ($\Phi_{\min}$). The initial configuration is selected in two steps: after 5000 years of model simulation, 5 candidates that compare best to present-day observations of ice geometry and speed (Fretwell et al., 2013; Rignot et al., 2011) are selected and continued. After $12,000$ years the best fit equilibrium result was selected among them and used as initial configuration for the projections, see Fig. S.1. We assess the ensemble members at each step using a scoring method (Pollard et al., 2016; Albrecht et al., 2020) that tests for root-mean-square deviation to present-day ice thickness, ice-stream velocities, as well as deviations in grounded and floating area, and the average distance to the observed grounding line position. We lay a specific focus on the Amundsen region, Filchner-Ronne and Ross ice shelves by additionally evaluating each indicator for these drainage basins individually.

The historic simulation is based on the same initial steady-state configuration and additionally applies atmospheric and oceanic forcing over the period from 1850 to 2014 as described below. The initial state for the experiments without historic simulation, hereafter referred to as INIT*, and the initial configuration after the historic simulation, hereafter referred to as INIT, are shown in Fig S.2. The INIT* configuration is very close to a steady-state with ice volume change rates being $5\,\mathrm{mm}$ over 85 years while the INIT state is out of balance with ice volume change rates being $-1.5\,\mathrm{cm}$ over 85 years, see Table 1. The INIT state in 2014 after the historic simulation scores very similar to the best-scoring initial configuration INIT*. For example, the root-mean square deviation in stream velocity in the Amundsen Sea region is $113\,\mathrm{m\,a^{-1}}$ for INIT (improved from $116\,\mathrm{m\,a^{-1}}$ for INIT*), in the Ross Sea $35\,\mathrm{m\,a^{-1}}$ (compared to $33\,\mathrm{m\,a^{-1}}$), in the Weddell Sea $47\,\mathrm{m\,a^{-1}}$ ($38\,\mathrm{m\,a^{-1}}$) and in the entire domain $290\,\mathrm{m\,a^{-1}}$ ($262\,\mathrm{m\,a^{-1}}$). The root-mean square deviation in grounded ice thickness is $166\,\mathrm{m}$ ($165\,\mathrm{m}$) in the Amundsen Sea, $188\,\mathrm{m}$ ($189\,\mathrm{m}$) in the Ross Sea, $167\,\mathrm{m}$ ($167\,\mathrm{m}$) in the Weddell Sea and $250\,\mathrm{m}$ ($250\,\mathrm{m}$) for the entire continent. The mean grounding line deviation is $12\,\mathrm{km}$ ($13\,\mathrm{km}$) in the Amundsen Sea, $24\,\mathrm{km}$ ($24\,\mathrm{km}$) in the Ross Sea, $14\,\mathrm{km}$ ($15\,\mathrm{km}$) in the Weddell Sea and $17\,\mathrm{km}$ ($17\,\mathrm{km}$) in the entire domain.

## 2.2 Experiments

We here present experiments based on the ISMIP6, LARMIP-2 and initMIP protocols that were done for both initial configurations. A list of all experiments is given in Table S.1. The initMIP experiments employ idealized forcing designed to test the model response to simplified forcing of the surface mass balance (experiment 'asmb'), and the basal mass balance (experiment 'abmb') which increase linearly for 50 years and are kept constant afterwards (Seroussi et al., 2019).

For LARMIP-2, constant step-forcing perturbations of the basal mass balance ($4, 8$ and $16\,\mathrm{m\,a^{-1}}$) are applied in five Antarctic regions (Antarctic Peninsula, East Antarctica, Ross Sea, Amundsen Sea, Weddell Sea). From the modeled sea-level response, linear response functions are derived that can be used to emulate the model's response to arbitrary melt forcing.

The ISMIP6 protocol prescribes atmospheric and oceanic forcing from CMIP5 models. We use the forcing data provided by ISMIP6 for (1) NorESM1-M for RCP8.5, (2) MIROC-ESM-CHEM for RCP8.5, (3) NorESM1-M for RCP2.6 and (4) CCSM4

for RCP8.5 (experiments 1-4 in Seroussi et al., under review). To be consistent with LARMIP-2, we here only apply the ocean forcing in projections and keep the surface mass balance constant.

We run experiments for both initial configurations with * indicating simulations starting from the pseudo-steady state in 2015, INIT*. The control experiments for both initial configurations employ constant climate conditions as described in the following two subsections.

## 2.3 Atmosphere forcing

Surface mass balance and ice surface temperatures for the initial configuration without historic forcing are from RACMOv2.3p2 (1986 to 2005 averages, Van Wessem et al., 2018), remapped from $27\,\mathrm{km}$ resolution. The historic simulation is started from the same conditions with historic surface mass balance and surface temperature changes following the NorESM1-M simulation as suggested by ISMIP6 (Bentsen et al., 2013). The historic forcing from NorESM1-M is normalized to its initial period (1950-1980) and the anomalies are then added to the constant climatology from RACMO. Since the provided data starts in 1950, surface mass balance and temperatures are constant between 1850 and 1949. Over that period, the aggregated, yearly surface mass balance is very similar to the RACMO climatology, as shown in Fig. S.3.

In contrast to ISMIP6, where surface mass balance and surface temperature changes are driven by GCM data, we here keep surface conditions - in line with LARMIP-2 - constant throughout the projections. Note that due to changes in the ice-sheet extent, surface mass balance integrated over the entire ice sheet might change slightly, see Table 1. Surface mass balance and temperatures in the projections that start from the pseudo steady-state INIT* are given by the RACMO climatology. For the projections based on the historic simulation we created a new climatology to account for increases in surface mass balance and temperatures in the historic simulation. We avoid using exceptionally high or low values that arise from interanual variability at a specific snapshot in time by using the 1995 to 2014 average of the respective fields.

## 2.4 Ocean forcing

Sub-shelf melt rates are calculated by PICO which extends the ocean box model by Olbers and Hellmer (2010) for application in 3-dimensional ice sheet models (Reese et al., 2018a). It mimics the vertical overturning circulation in ice-shelf cavities and has two model parameters that apply for all Antarctic ice shelves simultaneously: $C$ related to the strength of the overturning circulation and $\gamma_T$ related to the vertical heat exchange across the ice-ocean boundary layer. We here use parameters $C = 1 \times 10^6\,\mathrm{m^6\,s^{-1}\,kg^{-1}}$ and $\gamma_T = 3 \times 10^{-5}\,\mathrm{m\,s^{-1}}$ that were found to yield realistic melt rates in comparison to present-day estimates (Reese et al., 2018a; Rignot et al., 2013). The value of $\gamma_T$ is slightly higher than the reference value as an outcome of the ensemble study, see Fig. S.1.

We initialize PICO with an ocean data compilation from the World Ocean Atlas 2018 pre-release (Locarnini et al., 2018; Zweng et al., 2018) and Schmidtko et al. (2014). PICO is driven by ocean temperature and salinity averaged over the depth of the continental shelf within each drainage basin. The data from the WOA2018 pre-release is processed by determining the relevant depth from bathymetric access to ice-shelf cavities. In Dronning Maud Land (PICO basins 2 to 5), where ocean temperatures have a warm bias due to the lack of data along narrow continental shelves, values from Schmidtko et al. (2014)

were used. Using the currently observed 'warm' conditions in the Amundsen Sea, we found that region to collapse in the initial
ensemble irrespective of basal sliding parameters. As this region is out of balance today due to oceanic forcing (e.g., Konrad
et al., 2018; Shepherd et al., 2018), it would be inconsistent to initialize our model by running it towards equilibrium over
several thousand years applying constant present-day climate forcing. We hence reduced temperatures in the Amundsen Sea to
'cold' conditions ($-1.25\,°C$, Jenkins et al., 2018).

Ocean temperature and salinity forcing is calculated from CMIP5 models using an anomaly approach as suggested for ISMIP6
(Barthel et al., 2019; Jourdain et al., under review). We average these values over $400$ to $800\,m$ depth to obtain suitable input
for PICO. The historic forcing is based on NorESM1-M (as suggested for ISMIP6) and anomalies are normalized to the initial
period (here 1850-1900), similar to the atmosphere forcing. A new ocean climatology for the experiments starting from the
historic simulation is obtained from the 1995 to 2014 average conditions.

For LARMIP-2, we add melt rate anomalies to the underlying PICO melt rates in different Antarctic regions as described in
Levermann et al. (2020). Using linear response theory, the probability distribution of the sea-level contribution for RCP8.5 is
then calculated following the LARMIP-2 protocol.

## 3 Results

We present here (1) the results for the two initial configurations submitted to ISMIP6 and (2) the sea-level estimates for RCP8.5
obtained following the LARMIP-2 and ISMIP6 experiments based on the historic configuration.

### 3.1 Initial configurations and historic simulation

The two initial configurations for 2015, one based on a pseudo-equilibrium and one on a historic simulation from 1850 to 2014,
do not differ much in terms of state variables such as ice thickness, volume or speed (see Sect. 2.1). However, the configurations
have opposed change rates: INIT* has a small tendency to gain mass and INIT is clearly out of balance and loses mass (compare
the control simulations in Table 1). Over the historic period, the ice sheet thins along its margins through increased sub-shelf
melting and at the same time thickens in the interior due to more snowfall. These signals are smaller than $50\,m$ over grounded
regions, see Fig. S.2. The thinning of ice shelves around the margins and subsequent reduction of buttressing causes the ice
streams and ice shelves to slightly speed up over the historic simulation. The sensitivity of the modeled ice thickness and
velocities to the historic forcing is smaller than the sensitivity to different parameters in the initial ensemble, see Fig. S.1.
Overall, continent-wide aggregated basal mass balance decreases stronger than the aggregated surface mass balance increases,
leading to mass loss of $3.6\,mm\,SLE$ between 1850 and 2014 in comparison to the historic control simulation, see Fig. 1 and
Fig. S.3. This is smaller than the observed mass loss of $7.6 \pm 3.9\,mm\,SLE$ between 1992 and 2017 (Shepherd et al., 2018).
The patterns of present-day thickness changes (here 2014) are more realistic in the historic configuration INIT than for the
pseudo-equilibrium state INIT*. Furthermore, highest mass losses are simulated in the Amundsen Sea and Totten regions which
agrees with observations (Shepherd et al., 2018). Both initial configurations are further compared to other model configurations
and to present-day ice thickness and velocities in Seroussi et al. (under review).

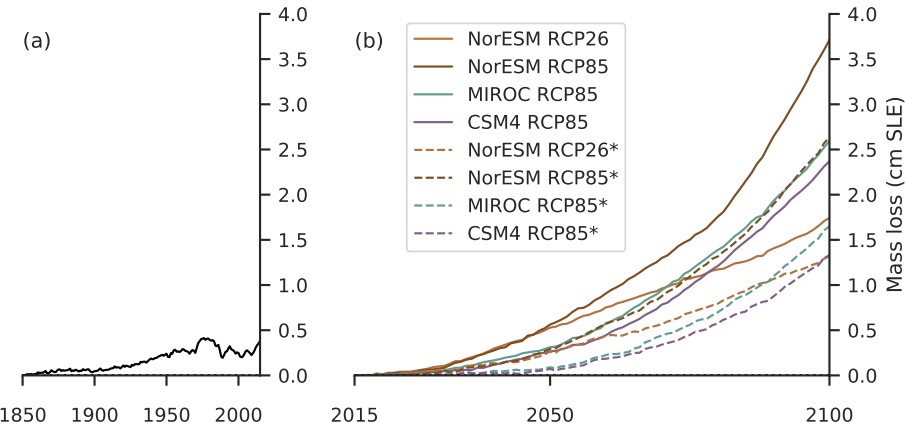

**Figure 1.** Historic simulation and projections of the Antarctic Ice Sheet driven by ISMIP6 ocean forcing. Shown is the evolution of the sea-level contribution (a) for the historic simulation relative to its control simulation and (b) for the projections with respect to the control simulations, in cm sea-level equivalent (SLE). Experiments are initialized either from a historic run (solid lines) or from the initial state (dashed lines) and forced with changes in ocean temperature and salinity from the ISMIP6 experiments no. 1 to 4 with the respective CMIP5 model indicated in the legend.

### 3.2 Comparison to initMIP Antarctica

Results from the idealized surface mass balance experiment 'asmb' as described in initMIP Antarctica (Seroussi et al., 2019) are very similar for both initial states with $119\,\mathrm{mm}\,\mathrm{SLE}$ of mass gains for the 'historic' configuration INIT and $118\,\mathrm{mm}\,\mathrm{SLE}$ for the 'cold-start' configuration INIT* after 85 years of simulation with respect to the control simulations, see Table 1. This is close to the response of the different models that participated in initMIP Antarctica which showed mass gains between 125 and $186\,\mathrm{mm}\,\mathrm{SLE}$ after 100 years.

For the idealized basal melt rate experiment 'abmb' from initMIP Antarctica, both states are also quite similar with mass loss of 43 and $40\,\mathrm{mm}\,\mathrm{SLE}$ after 85 years, respectively, see Table 1. In comparison, in Seroussi et al. (2019) a model spread of 13 to $427\ \mathrm{mm}\,\mathrm{SLE}$ after 100 years is reported. Results for both configurations presented here are close to the median of model results for both experiments tested in initMIP Antarctica.

### 3.3 ISMIP6 ocean-forcing experiments

We here compare simulations for both initial configurations that are driven by ocean forcing from the ISMIP6 experiments no. 1 to 4 (see Sect. 3.3 Seroussi et al., under review). In general, the ice sheet's mass loss increases with stronger ocean forcing as projected for RCP8.5 in comparison to RCP2.6, see Fig. 1. The highest losses for RCP8.5 are found for NorESM1-M. The magnitude of mass loss ranges from $1.4$ to $4.0\,\mathrm{cm}\,\mathrm{SLE}$ in comparison to the control simulation, which is substantially smaller than previous estimates of Antarctica's sea-level contribution (e.g., DeConto and Pollard, 2016; Golledge et al., 2019; Edwards et al., 2019) or expert judgement (Bamber et al., 2019). Furthermore, we find that the historic simulation makes

**Table 1.** Mass loss and evolution of surface and basal mass balance in ISMIP6 simulations. All values, except for the ctrl simulations, are relative to the respective control simulation.

| Experiments | $\Delta$SMB | $\Delta$BMB | $\Delta$SMB+$\Delta$BMB | Sea-level contribution |
|---|---|---|---|---|
| | $\mathrm{Gta}^{-1}$ | $\mathrm{Gta}^{-1}$ | $\mathrm{Gta}^{-1}$ | mm SLE |
| historic ctrl | 3 | 8 | 11 | -5.2 |
| historic | 65 | -428 | -362 | 3.6 |
| ctrl | -17 | 8 | -9 | 14.9 |
| asmb | 764 | -28 | 735 | -119.3 |
| abmb | -51 | -538 | -590 | 42.7 |
| NorESM RCP85 | -41 | -1071 | -1112 | 39.6 |
| MIROC RCP85 | -24 | -748 | -772 | 27.6 |
| NorESM RCP26 | -23 | -107 | -130 | 18.7 |
| CCSM4 RCP85 | -31 | -790 | -821 | 25.3 |
| ctrl* | 4 | 19 | 23 | -4.9 |
| asmb* | 770 | -25 | 746 | -117.5 |
| abmb* | -56 | -562 | -618 | 39.8 |
| NorESM RCP85* | -40 | -1024 | -1064 | 27.1 |
| MIROC RCP85* | -30 | -778 | -808 | 17.0 |
| NorESM RCP26* | -25 | -79 | -104 | 13.5 |
| CCSM4 RCP85* | -28 | -787 | -815 | 13.7 |

Experiments without the historic run are indicated by *. Changes in basal and surface mass balance from the first to the last time step in the experiments (i.e., from 1850 to 2014 in the historic run and from 2015 to 2100 in the other experiments).

the configuration more susceptible to ocean forcing, see Fig. 2. Ocean-driven mass loss in comparison to the control run is increased by about $50\%$ (factor 1.5) when starting from the historic simulation in contrast to the 'cold-start'.

## 3.4 LARMIP-2 basal melt rate forcing experiments

In LARMIP-2, sea-level probability distributions from the Antarctic Ice Sheet are derived using linear response functions as described in Levermann et al. (2020). The response functions are derived from experiments in which constant basal melt rate forcing is applied for five different regions of Antarctica. We here perform the same experiments for both configurations described in Sect. 2.

We find that for all regions the ice sheet response compares with the responses found in LARMIP-2 as, for example, in the PISM-PIK contribution that is based on a different initial state with $4\,\mathrm{km}$ horizontal resolution and that does not apply subgrid melting, compare Fig. 3 with Fig. 4 from Levermann et al. (2020). A detailed comparison of both PISM-PIK contributions is

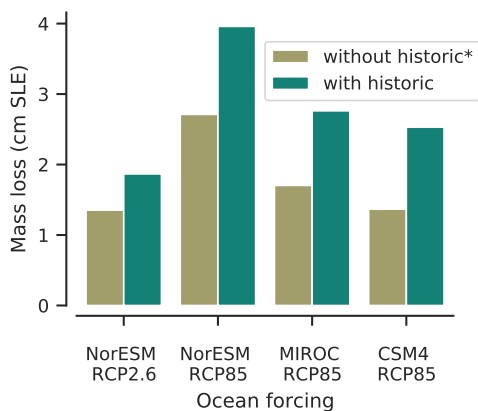

**Figure 2.** A preceding historic simulation increases the susceptibility of the ice sheet to ocean forcing in projections. Shown is the mass loss in simulations started directly from the initial state compared to simulations based on a historic run. The mass loss in 2100 is given with respect to the control simulation, after 85 years of applying the ocean forcing from ISMIP6 experiments no. 1 to 4 with the respective emission scenario / CMIP5 model indicated on the x-axis.

**Table 2.** Percentiles of the probability distribution of the sea level contribution from Antarctica under the RCP-8.5 climate scenario from 2015 to 2100, estimated following the LARMIP-2 protocol.

| percentile | cm SLE (INIT*) | cm SLE (INIT) | difference (%) |
|:---:|:---:|:---:|:---:|
| 5.0 % | 3.3 | 3.5 | 5.5 |
| 16.6 % | 8.5 | 9.1 | 6.8 |
| 50.0 % | 17.2 | 18.3 | 6.4 |
| 83.3 % | 33.9 | 35.8 | 5.7 |
| 95.0 % | 52.8 | 55.6 | 5.3 |

given in Table S.2. Similar to the ISMIP6 simulations, the experiments show different responses for the two initial configurations, especially in the Weddell Sea, East Antarctica and the Amundsen Sea region. The overall difference is smaller than in the ISMIP6 experiments for the stronger forcing applied here.

Following the procedure in LARMIP-2, we derive response functions from the idealized experiments for the five Antarctic regions. We then convolve the response function with basal melt rate forcing, given in Fig. 4, to obtain a probability distribution of the future sea-level contribution for RCP8.5 which is given in Fig. 5. The ocean-driven mass loss from 2015 to 2100 has a very likely range of 3.5 to 55.6 cm SLE, a likely range of 9.1 to 35.8 cm SLE and a median of 18.3 cm SLE (5 to 95%, 16.6 to 83.3%, and 50% percentiles, respectively, see Table 2). Similar to the ISMIP6 simulations, these results obtained for the historic initial configuration are larger than the results for the steady-state configuration, with increases between 5 and 7%. In comparison, the PISM-PIK contribution of LARMIP-2 has a very likely range of 7 to 48 cm SLE, a likely range of 11 to

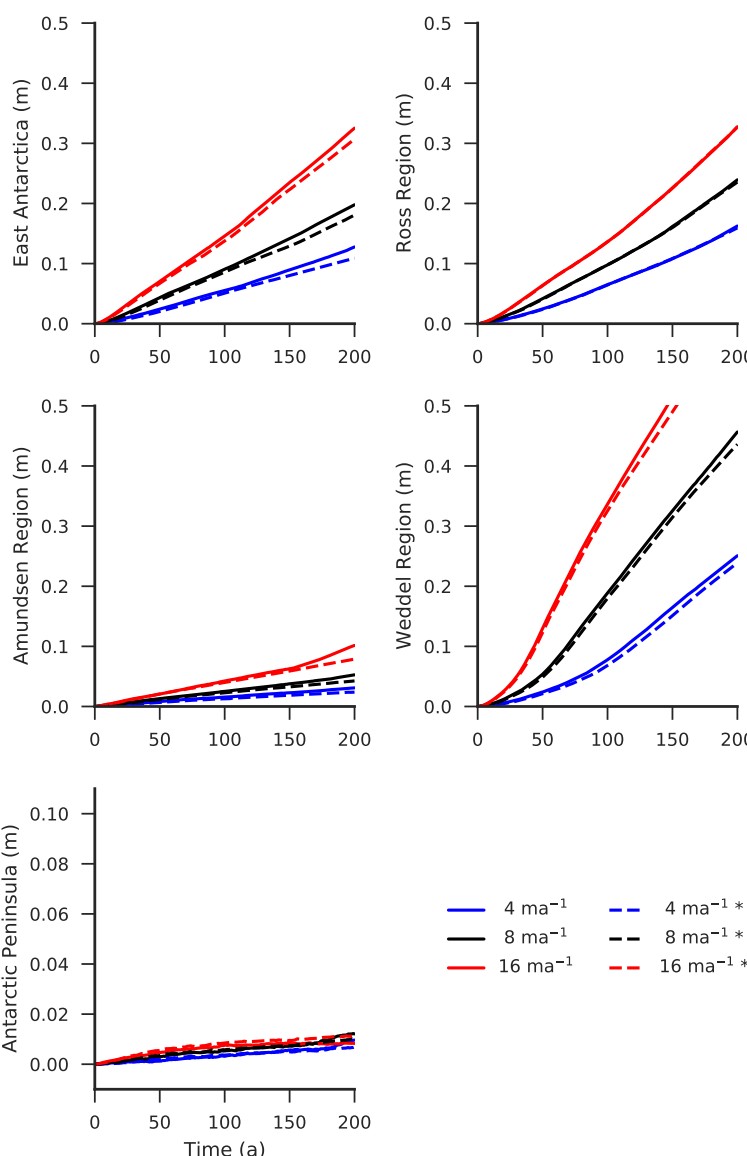

**Figure 3.** Mass loss of the different regions in Antarctica (indicated on y-axis) driven by constant LARMIP-2 basal melt rate forcing. For the experiments from the LARMIP-2 protocol we show the changes in volume above flotation initialized from a historic simulation (solid line) and from the initial state directly (dashed line, indicated by *). Mass loss is shown relative to the respective control simulation. From the response of the ice sheet to a constant melt rate forcing over 200 years, a response function is derived which serves then to emulate the sea-level contribution under various climate scenarios. This figure is similar to Fig. 4 in Levermann et al. (2020).

31 cm SLE and a median of 19 cm SLE for the 21st century. The resulting sea-level probability distribution is hence in line with the estimates presented in LARMIP-2.

 **4 Discussion**

In the following, we compare the results found in the ISMIP6 and LARMIP-2 experiments, discuss the role of the ocean forcing and of the historic simulation.

**4.1 Comparison of LARMIP-2 and ISMIP6 sea-level projections**

The projected mass loss in LARMIP-2 is an order of magnitude larger than the ocean-driven mass loss in our ISMIP6 exper-
 iments for RCP8.5, see Sect. 3. In order to understand this difference better, we here investigate the ocean forcing in more detail.

Both intercomparison projects, ISMIP6 and LARMIP-2, are based on CMIP5 model sub-surface ocean temperature changes (Levermann et al., 2014; Barthel et al., 2019; Jourdain et al., under review). While they are directly applied in ISMIP6, they are used to derive a scaling between global mean temperatures and Antarctic sub-surface temperatures in LARMIP-2. While minor
 differences in ocean forcing might occur due to different processing steps, a more significant difference is that the LARMIP-2 experiments are driven by basal melt rate changes emulated from the forcing, while in the ISMIP6 simulations ocean forcing is translated into basal melt rates via sub-shelf melt parameterizations, in our case PICO.

Figure 4 shows projected basal melt rates and their uncertainty ranges for RCP8.5 used in LARMIP-2 together with the basal melt rate changes in the ISMIP6 simulations. Note that LARMIP-2 assumes constant changes in basal melt rates over the
 entire ice shelf. In contrast, since PICO mimics the vertical overturning circulation in ice-shelf cavities, basal melt rates in the ISMIP6 simulations increase strongest along the grounding line (in PICO's first box) and less towards the ice shelf front. The melt rate changes in PICO along the grounding line are hence an upper limit for the comparison to the LARMIP-2 forcing while the shelf-wide averaged changes provide a lower limit. Overall, we find that in the ISMIP6 simulations, basal-melt rates increase more in regions with smaller ice shelves than in the Ross and Weddell Sea. Furthermore, we find that the basal melt
 rate changes in our ISMIP6 contribution in all Antarctic regions is located in the lower range (percentiles) of the LARMIP-2 forcing. Only for the Antarctic Peninsula, PICO melt rates along the grounding line increase stronger than the median in LARMIP-2 for NorESM1-M and MIROC. For all other regions, melt rate changes along the grounding line are smaller than the median in LARMIP-2 ($50\%$-percentile). For the Amundsen Sea region, they lie within the likely range ($16.6\%$ to $83.3\%$ percentiles), for East Antarctica and the Ross Sea, they are around the lower margin of the likely range and for the Weddell
 Sea, they are lower than the very likely range ($5\%$ to $95\%$ percentiles). Shelf-wide changes are generally smaller than the likely range, for the Weddell Sea and the Antarctic Peninsula they are also smaller than the very likely range.

This is consistent with the mass loss in the ISMIP6 simulations being lower than the likely range of LARMIP-2 for almost all regions, see Fig. 5. These findings are underlined by the direct comparison with the PISM-PIK contribution to LARMIP-2 which is based on a different initial setup, see Sect. 3.4. Note that basal melt rate changes in East Antarctica seem similar
 in Fig. 4 for NorESM1-M and MIROC but mass loss is higher for NorESM1-M, because the ocean forcing in the ISMIP6 simulations varies across the different ice shelves in East Antarctica. While there is stronger ocean warming in Dronning Maud Land and Amery in the MIROC forcing, the ocean warms substantially more in the Totten region for NorESM1-M. The higher

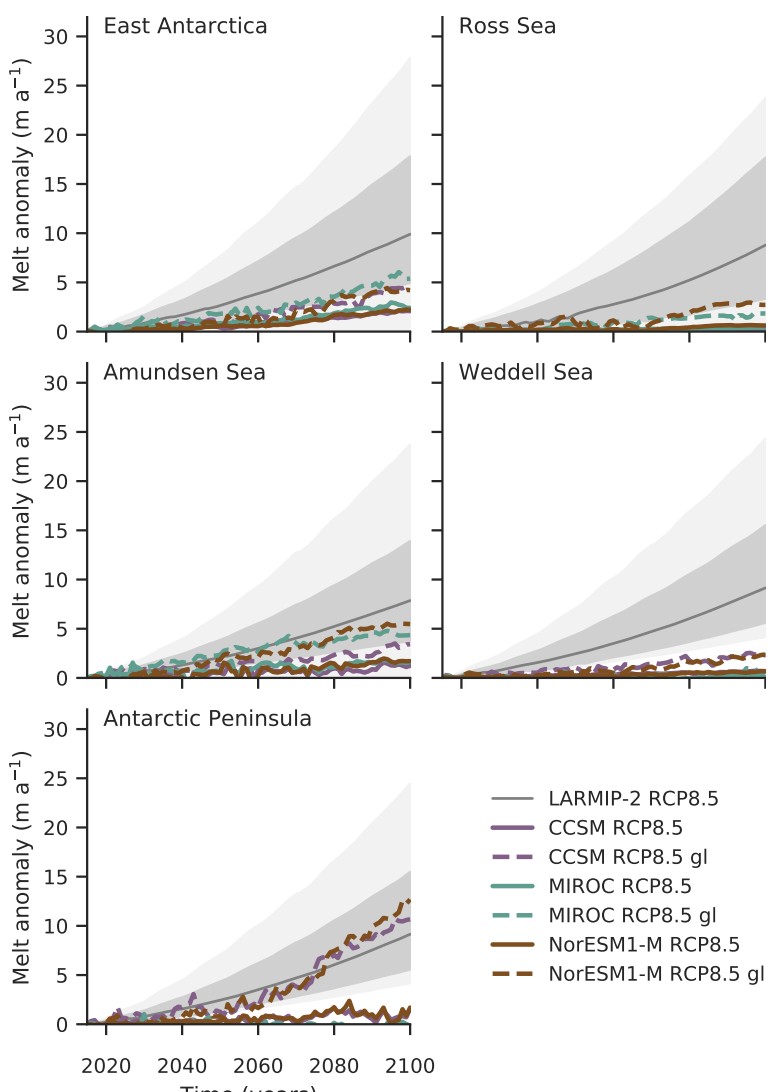

**Figure 4.** Projected basal melt rate changes in the different Antarctic regions from LARMIP-2 and in the ISMIP6 contribution forced with NorESM1-M, CCSM and MIROC ocean changes under RCP8.5. In LARMIP-2 spatially constant basal melt rate forcing is applied with corresponding very-likely ranges (5 to 95%-percentiles, light gray shading), likely ranges (66%-percentile around the median, dark gray shading) and median (gray line) for the RCP8.5 scenario. In the ISMIP6 contribution, basal melt rates are calculated by PICO, which shows higher increases close to the grounding line (PICO Box 1, indicated by 'gl') than averaged over the ice shelves. Figure is similar to Fig. 3 in Levermann et al. (2020).

vulnerability of the Totten region then causes higher overall mass loss.

In Figure 6 we assess for each region the mass loss by applying the response functions to the corresponding PICO melt rate
260   changes driven by NorESM1-M ocean forcing, once averaged over the entire ice shelves and once close to the grounding lines.

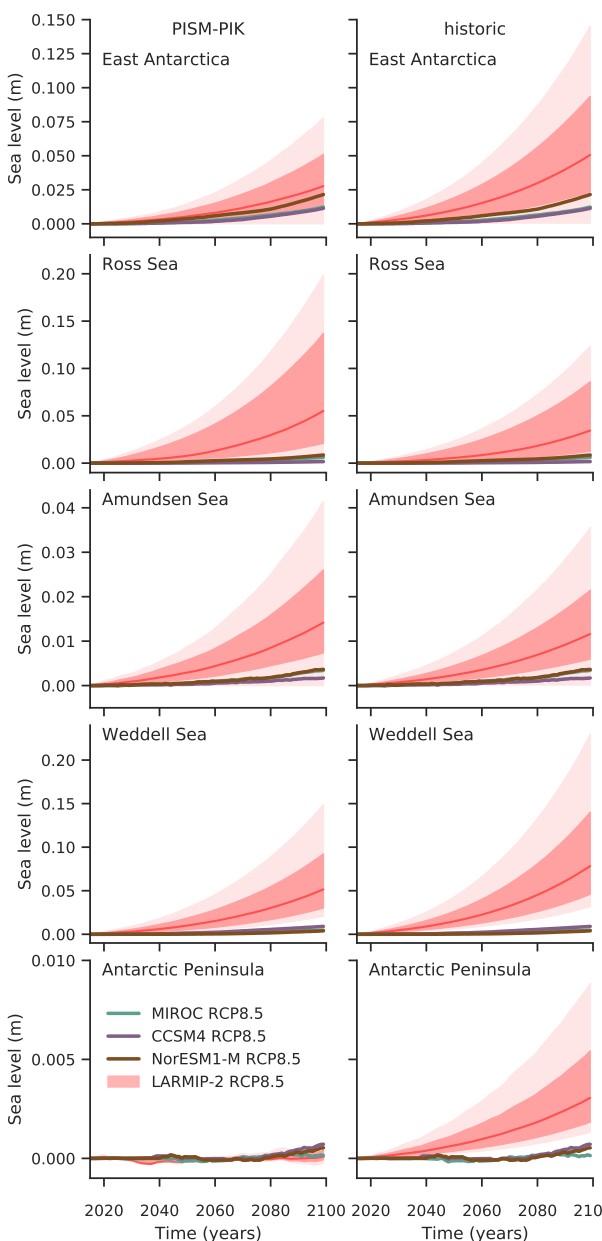

**Figure 5.** Projections of Antarctica's sea level contribution under the RCP8.5 climate scenario for the different Antarctic regions for LARMIP-2 and for the ISMIP6 experiments driven by NorESM1-M, MIROC and CCSM4 ocean forcing. The very likely ranges (5 to 95%-percentiles, light red shading), likely ranges (16.6 to 83.3%-percentiles, dark red shading) and the respective median (50%-percentile, red lines) of mass loss is shown for (left panels) the PISM-PIK simulations submitted to LARMIP-2 and, for comparison, (right panels) estimated following LARMIP-2 for the setup as submitted to ISMIP6.

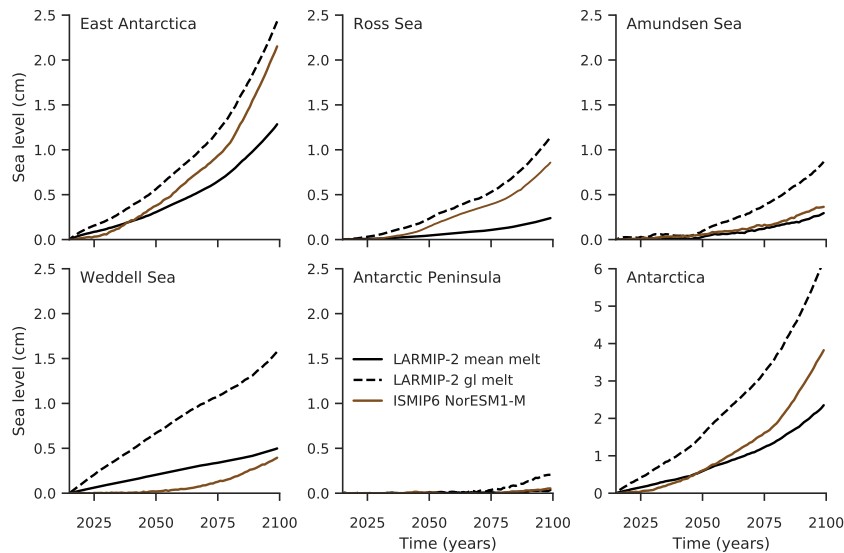

**Figure 6.** Projections using PICO forced with NorESM1-M ocean conditions compared to projection obtained by the response function. The response function is derived for the INIT configuration. It is applied to the basal melt rate forcing from PICO using average conditions underneath the shelves in the corresponding sector (generally an underestimation) and using the melting at the grounding line (generally an overestimation) from Fig. 4.

When comparing the respective mass loss with the ISMIP6 simulation, we find that indeed the changes at the grounding line provide an upper limit while the changes over the entire ice shelf provide a lower limit for the actual mass loss.

Overall, we find that mass losses in the ISMIP6 projections are generally lower than the likely range in LARMIP-2, and in the Weddell Sea losses are smaller than the very likely range, as the basal melt rate changes in the LARMIP experiments are an order of magnitude higher than those estimated with PICO and ISMIP6 forcing.

## 4.2 Role of ocean forcing and basal melt rate sensitivity

In order to gain a better understanding of the conversion of ocean forcing to basal melt rates in LARMIP-2 and in our ISMIP6 contribution, we further analyse the sensitivity to ocean warming.

We perform step-forcing experiments for both initial configurations and diagnose the effect on basal melt rates, see Fig. 7. Ocean temperatures are increased by $0.5, 1, 2, 3$ and $4\,°\mathrm{C}$ and the corresponding basal melt rates for constant ice-shelf geometries are diagnosed using PICO. We find that the sensitivity in the Amundsen Sea Region is comparatively high with about $10\,\mathrm{ma^{-1}K^{-1}}$, while the sensitivity in the Weddell Sea is lower with about $1.5\,\mathrm{ma^{-1}K^{-1}}$, which yields for the entire Antarctic ice shelves an overall sensitivity of about $2.2\,\mathrm{ma^{-1}K^{-1}}$. The sensitivities for melting close to the grounding line are as expected a bit higher: $10.5\,\mathrm{ma^{-1}K^{-1}}$ for the Amundsen Sea region, $3.9\,\mathrm{ma^{-1}K^{-1}}$ for the Weddell Sea and $5.3\,\mathrm{ma^{-1}K^{-1}}$ on average for all Antarctic ice shelves. In both cases, the Antarctic-wide sensitivity is substantially lower than the sensitivity used in LARMIP-2. In the latter study, a sensitivity between 7 and $16\,\mathrm{ma^{-1}K^{-1}}$, based on Jenkins (1991) and Payne et al. (2007),

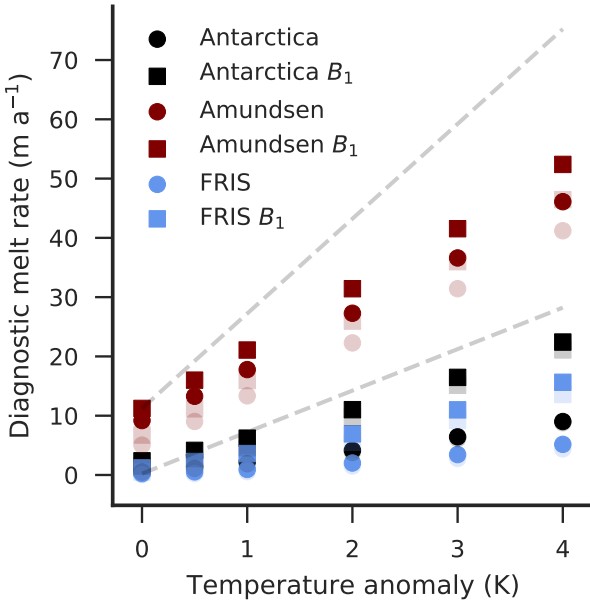

**Figure 7.** Sensitivity of basal melt rates to ocean temperatures in PICO. Diagnosed from the historic configuration (opaque) and the cold-start configuration (transparent) in 2015 using step-wise ocean temperature increases. Dots show shelf-wide averages while boxes indicate the basal melt rates close to the grounding lines (in PICO box $B_1$). The dashed grey lines indicate the sensitivity estimates used in Levermann et al. (2020).

is assumed to translate ocean forcing into sub-shelf melt rates. This is consistent with our findings in the previous section that in the ISMIP6 simulations mass loss from the Antarctic Ice Sheet, and especially from the regions that drain the large Filchner-Ronne and Ross ice shelves, is smaller than the likely range estimated following the LARMIP-2 protocol. Jourdain et al. (under review) report that a different tuning of the ISMIP6 basal melt parameterization to fit observations in the Amundsen Sea (from Dutrieux et al., 2014; Jenkins et al., 2018) substantially increases the sensitivity to ocean changes and Seroussi et al. (under review) find that this enhances the sea-level contribution by a factor of six.

Since the sensitivity in PICO depends on the parameters used, with the overturning coefficient $C$ affecting the sensitivity in large ice shelves and the heat exchange coefficient $\gamma_T$ affecting the sensitivity in small ice shelves, a different tuning could improve basal melt rate sensitivities and thereby lead to higher mass losses in the ISMIP6-experiments. A consistency of sub-shelf melt rates with present-day observations could be achieved by introducing additional degrees of freedom through temperature corrections that reflect uncertainties in ocean properties, as for example used in Lazeroms et al. (2018) and Jourdain et al. (under review). In addition, tuning to realistic melt rates close to the grounding lines (in PICO's first box) is potentially more important than fitting shelf-wide melt rates (e.g., Goldberg et al., 2019; Reese et al., 2018b).

Few observations exist for targeted tuning of the sensitivity of basal melt rate parameterizations to ocean temperatures, hence the use of dynamic modelling of the ocean circulation in ice-shelf cavities could be explored. We estimate that the sensitivity

in Seroussi et al. (2017) varies between 6 and $16\,\mathrm{ma^{-1}K^{-1}}$ with an average of $9.4\,\mathrm{ma^{-1}K^{-1}}$ over the first twenty years of model simulation, which would be in line with the sensitivities used in LARMIP-2, see Fig. S.5.

Note that we provide linear estimates of the sensitivity of PICO in the discussion above, while Holland et al. (2008) report a quadratic dependency of melt rates on thermal forcing. They also discuss that the sensitivity depends on ice shelf cavity properties such as the slope of the ice-shelf draft and shape of the ice shelf and that sensitivities are generally higher close to the grounding line. Further factors that influence ocean circulation, such as bathymetry, also affect the ocean sensitivity. While PICO takes into account, that not all heat content of the ocean water masses that enter the cavity might be used for melting, it does not capture three-dimensional circulations in ice-shelf cavities that play a role especially for larger ice shelves such as Filchner-Ronne.

## 4.3 Role of historic trajectory of the Antarctic Ice Sheet

We find that while the historic simulation has no large effect on the initial sea-level volume (the overall difference being $1.6\,\mathrm{mm}$ SLE), it affects the mass loss in the projections. A number of reasons might cause the simulations starting from the historic configuration (INIT) to be more vulnerable to ocean forcing: both simulations have different initial trends of the sea-level relevant volume and rates of ice thickness change. These trends, or the different geometry after the historic simulation, might make the configuration more susceptible to ocean forcing, for example through non-linear changes in ice-shelf buttressing. In addition, the historic simulation might have pushed the ice sheet (closer) to a local instability that evolves in the projections. Figure S.6 shows the mass loss rates for all simulations presented in Sect. 3.3. In general, the rates are higher in the simulations based on the historic configuration, and these differences increase over time. In the RCP8.5 simulation forced with NorESM1-M ocean conditions, at around year 2075 a clear shift to substantially higher differences is visible. We hypothesize that this could be linked to a local instability that is kicked-off for the simulations starting from the historic configuration but not for those starting from the pseudo steady state. This is less pronounced for CCSM4 and MIROC, maybe due to differences in the ocean forcing and regions contributing to sea level rise. In the idealized experiments for LARMIP-2 (Fig. 3), differences in simulations starting from the two initial states arise especially in East Antarctica, the Weddell Sea and the Amundsen Sea, less in the Ross Sea. The effect of the historic simulation is reduced for the stronger basal melt rate forcing applied in the LARMIP-2 experiments, with mass loss increases in the projections between 5 and 7%.

Furthermore, the ice sheet's response might have changed after the historic simulation due to changes in boundary conditions. Moreover, changes in the ice-sheet state could result since, for instance, the underlying equation system depends non-linearly on the three-dimensional temperature field. The grounding line retreats during the historic simulation slightly into deeper regions, where the local freezing point at the ice shelf base near the grounding line is decreased due to its pressure dependence. Hence more heat is available for melting the ice-shelf base, which shows also in an increased sensitivity to ocean changes, see Fig. 7. In particular for lower temperatures, PICO shows a non-linear sensitivity of melt rates to ocean temperatures, as discussed in Reese et al. (2018a). Further investigations would be required to disentangle the reasons for the increased susceptibility to ocean warming after the historic simulation, also considering the strength of the forcing applied.

The sea-level contribution over the historic period from 1850 to 2014 is $3.6\,\mathrm{mm}$ in comparison to the control simulation. This

is smaller than the reported mass loss of the Antarctic Ice Sheet that amounts to $7.6 \pm 3.0\,\mathrm{mm}$ SLE between 1992 and 2017 (Shepherd et al., 2018). An improved understanding of the basal melt rate sensitivity, potential biases in the atmospheric or oceanic forcing, as well as an extension of the scoring with observed patterns of thickness changes would allow for performing 'hindcasting' experiments that, in a next step, could inform future projections.

## 5 Conclusions

In this study we compare sea-level projections for RCP8.5 from the Antarctic Ice Sheet as submitted to ISMIP6, using the PICO basal melt rate parameterization and constant surface mass balance forcing, and projections derived following the LARMIP-2 protocol that scales global temperatures to subsurface temperatures and melt rates, both using the Parallel Ice Sheet Model. Overall, we find that the sea-level contribution driven by ocean forcing in ISMIP6 is smaller than the likely range of the sea-level probability distribution in LARMIP-2. This difference can be explained by the comparably low sensitivity of melt rates to ocean temperature changes for the parameter tuning in PICO in comparison to LARMIP-2 where a sensitivity of 9 to $16\mathrm{m/a/K}$ is used that we found to be consistent with a coupled simulation of Thwaites glacier (Seroussi et al., 2017). Future sea-level projections should hence carefully consider the sensitivity of basal melt rates to ocean changes. Additional observations of ocean conditions and ocean-induced melt rates in combination with ocean modelling are needed to better constrain this sensitivity for the diverse ice-shelf cavities in Antarctica. Furthermore, we find that while the initial state resulting from a historic simulation from 1850 to 2014 is virtually indistinguishable from a steady-state simulation, the historic simulation increases the projected mass loss in 2100 by up to 50%. This means that not only the currently committed sea-level contribution should be considered in projections, but also the effect of the historic forcing on the ice sheet's susceptibility to ocean changes. 'Hindcasting' experiments, that reproduce observed thinning rates and ice loss over the past decades, would be valuable to better constrain model parameters and improve confidence in projections. Hence, further investigations are needed to assess the sensitivity of basal melting to ocean temperatures for basal-melt parameterizations and the role of historical forcing and initial conditions in future sea-level projections.

*Code and data availability.* Data and code is available from the authors upon request. Model outputs from the simulations for ISMIP6 described in this paper will be made available via the ISMIP6 project with digital object identifier. The PISM code as well as the scripts to analyse the simulations and create the figures will be made available with digital object identifier reference. The processing of the World Ocean Atlas pre-release data is described in the Bachelor thesis by Lena Nicola (2019) and shared upon request.

*Competing interests.* Helene Seroussi is an editor of the special issue The Ice Sheet Model Intercomparison Project for CMIP6 (ISMIP6). The authors declare that no other competing interests are present.

*Acknowledgements.* We thank the Climate and Cryosphere (CliC) effort, which provided support for ISMIP6 through sponsoring of workshops, hosting the ISMIP6 website and wiki, and promoted ISMIP6. We acknowledge the World Climate Research Programme, which, through its Working Group on Coupled Modelling, coordinated and promoted CMIP5. We thank the climate modeling groups for producing and making available their model output, the Earth System Grid Federation (ESGF) for archiving the CMIP data and providing access, the University at Buffalo for ISMIP6 data distribution and upload, and the multiple funding agencies who support CMIP5 and ESGF. We thank the ISMIP6 steering committee, the ISMIP6 model selection group and ISMIP6 dataset preparation group for their continuous engagement in defining ISMIP6. This is ISMIP6 contribution No X. Development of PISM is supported by NASA grant NNX17AG65G and NSF grants PLR-1603799 and PLR-1644277. The authors gratefully acknowledge the European Regional Development Fund (ERDF), the German Federal Ministry of Education and Research and the Land Brandenburg for supporting this project by providing resources on the high performance computer system at the Potsdam Institute for Climate Impact Research. Computer resources for this project have been also provided by the Gauss Centre for Supercomputing/Leibniz Supercomputing Centre (www.lrz.de) under Project-ID pr94ga and pn69ru. R.R. is supported by the Deutsche Forschungsgemeinschaft (DFG) by grant WI4556/3-1 and through the TiPACCs project that receives funding from the European Union's Horizon 2020 research and innovation programme under grant agreement no. 820575. T.A. is supported by the Deutsche Forschungsgemeinschaft (DFG) in the framework of the priority program "Antarctic Research with comparative investigations in Arctic ice areas" by grant WI4556/2-1 and WI4556/4-1 and in the framework of the PalMod project (FKZ: 01LP1925D), supported by the German Federal Ministry of Education and Research (BMBF) as a Research for Sustainability initiative (FONA). H.S. was supported by grants from NASA Cryospheric Science and Modeling, Analysis, Predictions Programs.

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
