# Peer review of "The role of history and strength of the oceanic forcing in sea-level projections from Antarctica with the Parallel Ice Sheet Model"

_The Cryosphere, 2019_

## Referee Comment (RC1) · Anonymous Referee #1 · 21 Feb 2020

———- General comments:

This paper describes the sensitivity of ice-sheet projections to the initialization method (simulating the 1850-2014 historical period vs starting in 2015, after the same long spin-up in both cases). It also describes the sensitivity to ocean warming, and compares the effect of parameterised melting (through PICO in ISMIP6 simulations) to melt perturbations imposed in LARMIP2. These sensitivities are expressed as Antarctic contributions to sea level rise.

The paper is well written, although a few clarifications are required in the Methods section. While I find the results useful for the ice-sheet/climate community, I have a few

major comments that should probably be addressed before publishing this paper:

1- An important conclusion of this paper appears to be that ice-sheet models should simulate the entire historical period to make meaningful projections. However, this would not be true is the initial state was selected in a way to get present-day thinning rates (e.g. including an observational ice thinning rate in the score used to pick the initial state in the long-term ensemble). I don't expect the authors to change this in their methodology, but this should be discussed.

2- A strong ad hoc temperature correction is applied to the Amundsen Sea, with deep-ocean temperatures set to -1.25°C, i.e. relatively cold conditions. Then, it is claimed that the PICO tuning parameters cannot be changed because they were tuned to match observational ice-shelf averaged melt rates (in Reese et al. 2018 with other ocean forcing dataset). These aspects need to be investigated in this paper. Melting (and ideally the ice-sheet response) without temperature correction should be described, and other tuning parameters should be considered. Further, it could be argued that the important observational target in terms of ice-shelf buttressing is not the ice-shelf average but the average in PICO's first box (near grounding line).

3- Melt rates obtained through the ISMIP6 framework are compared to the melt rates imposed in LARMIP2 (based on previous plume-model studies). It would be interesting to describe how these two types of melt rates compared to other observational or modelling studies (see specific comment below).

———- Specific comments:

- L. 43-45: These are global feedbacks related to freshwater injection into the ocean. There are also important local thickness-melting feedbacks described by Timmermann and Goeller (2017) and Donat-Magnin et al. (2017).

- L. 51: "haven" -> "have".

- Section 2.1: I don't necessarily ask the authors to do it, but it would be interesting to

see the score of the historical simulation in 2015 compared to the ensemble of initial states and in particular the one that was selected against present-day observations of ice geometry and speed.

- L. 81: About "scoring against present-day observations of ice geometry": including the ice thinning rate in the score would have helped get an initial state similar to observations and maybe to the historic state at 2015.

- L. 89-90: What constant value is used for 1850-1949?

- L. 90-91: This sentence is unclear.

- Section 2.2: Specify here what is done for SMB in LARMIP-2.

- L. 96-97: "C=106" would be better than programming notation. And both C and $\gamma$T should have units. The $\gamma$T value does not seem to be exactly the same as Reese et al. (2018), why has it been modified despite a careful tuning in that paper?

- L. 98-99: How was the data compilation based on World Ocean Atlas (WOA) 2018 and Schmidtko's dataset built? Was the latest update of WOA 2018 used? If not it is worth being mentioned because there were important updates near Antarctica. See https://www.nodc.noaa.gov/OC5/woa18/woa18data.html

- L. 99-100: Changing the Amundsen Sea temperatures to -1.25°C is quite an issue. Does it mean that PICO produces overly high melt rates in the Amundsen Sea, or that PISM cannot handle melt rates close to observational estimates? In Levermann et al. (2020), a similar correction is done to -0.37°C with a claim that this possibly represents pre-industrial conditions. This needs to be discussed, and what's happening for uncorrected temperatures needs to be shown in this paper. It is a crucial point because (1) the Amundsen Sea is our best present-day test for future warm conditions, and (2) the small sensitivity reported in this paper with the ISMIP6 framework could be due to this artificial cooling in the Amundsen Sea.

- L. 103: Is NorESM1-M used from1850 onward? I don't understand what is the "new

climatology" and how it is used.

- L. 104: Not clear, what "ISMIP6 ocean forcing" refers to? It has just been explained that PICO is used, "initialized with an ocean data compilation from Locarnini et al. (2018) and Schmidtko et al. (2014)", with additional CMIP5 anomalies. Is there anything else to add?

- Arriving at section 3, it is surprising to see several CMIP5 models while only NorESM1-M was mentioned in section 2. The use of several models (all starting from NorESM1-M's 1850-2014 period) should be described in section 2. The methods should be understandable without knowing the full ISMIP6 framework.

- L. 124: "basal mass balance increased" or basal mass balance DEcreased? Same in Tab. 1, should SMB and BMB have the same sign for ice-sheet mass loss? Maybe just a matter of taste. . .

- Section 3.1: it might be clearer to define the "ctrl" experiment in section 2, and to explain how it is designed: which part of the historical forcing is kept constant until 2100?

- Tab. 1 and section 3.2: please briefly define "asmb" and "abmb" so that it can be understood without having to read Seroussi et al. (2019).

- Throughout section 3: In view of the drifts in ctrl vs ctrl*, it is expected to see different mass loss for the two initialisations. All the plots and mass loss estimates could be calculated with respect to ctrl or ctrl*. . .

- L. 146: you could cite Edwards et al. (2019) here as it revisits several previous estimates.

- L.147: it is difficult to visualise the 50%. You could consider redo Fig. 2 with ctrl and ctrl* subtracted.

- Section 3.4: The differences between the initial state used for PISM-PIK in Levermann

et al. (2020) and the one used for LARMIP-2 PISM simulations should be summarised somewhere in this manuscript. Without this it is difficult to understand what to expect from this section.

- Section 4.1: here only NorESM1-M results are shown because it "shows highest mass losses of all ISMIP6 experiments". This is true for the total Antarctic mass loss, but the results considering individual sectors (as in Fig. 4) may be higher in other models. Furthermore, LARMIP-2 is based on a CMIP5 19-model mean, so it seems more appropriate to use the ISMIP6 multi-model mean here rather than just NorESM1-M.

- L.224-225: It is true that tuning C and $\gamma$T would "affect the comparison of sub-shelf melt rates to present-day observations". But (1) the input temperatures have been modified compared to the initial tuning of PICO, and (2) it could also be argued that the important observational target is not the ice-shelf average but the average in PICO's box 1 (near grounding line). In view of Reese et al. (2018b), this is what matters the most for buttressing, isn't it?

- Section 4.2: The sensitivity used in LARMIP (7 to 16 m/yr/K) was estimated from plume models (Jenkins 1991 and Payne et al. 2007). It is discussed in the manuscript as if these were well-established values. These numbers may be acceptable, but there have been a few observational and more complex modelling studies since then, which have estimated sensitivities to ocean warming. It would be useful to mention whether these numbers can still be considered as plausible (Naughten et al. 2018; Séroussi et al. 2017; Jenkins et al. 2018). Besides, it is possible that the ocean sensitivity depends on the ice shelf under consideration (for example because ocean heat entering a very large ice-shelf cavity will be entirely consumed, while only a part of the available heat is consumed in smaller cavities, which may be captured by PICO?). Please also discuss these aspects.

- L.225-227: a similar correction has actually been applied to the Amundsen Sea in

this paper.

---

## Referee Comment (RC2) · Anonymous Referee #2 · 6 Apr 2020

**Summary**

In *The role of history and strength of the oceanic forcing in sea-level projection from Antarctica with the Parallel Ice Sheet Model*, Reese et al. present the results of applying a suite of numerical experiments associated with both the ISMIP6 and LARMIP-2 model intercomparisons to the parallel ice sheet model. The key findings of this paper are that 1) the inclusion of a century-scale climate history leads to a significantly different mass loss trajectory for all of the experiments included in ISMIP6 (in particular, a climate history leads to greater mass loss), and 2) that the mechanism for parameterizing sub-shelf melting matters alot, and that choosing between two different methods

leads to an order of magnitude difference in century scale sea-level predictions.

**General comments**

Overall, I find the paper to be well-written and very interesting. The scientific quality is high, and the conclusions presented will be useful for those trying to hone in on areas of remaining uncertainty when prognosticating with regard to Antarctica. One overarching comment is that the experiments that are applied should be described more fully: because this paper deals with numerical experiments drawn from two other works (the main ISMIP6 forcing paper and the LARMIP-2 paper, one of which is still in review), it would be helpful to briefly state the assumptions and differences between them (to aid in intercomparing the intercomparisons, as it were). My remaining comments are on a line-by-line basis, and may be found below.

**Specific comments**

**L80** A description of the mechanism for creating the ensemble, as well as the scoring method, would be appropriate here. Additionally, a discussion of the degree to which the optimal ensemble member actually matched observations would help in determining how seriously the reader should take the predictions included in this work.

**L86** Add "forcing" after "historic".

**L89** Please add a citation for NorESM1-M.

**L89–90** Please describe how the climate constants were selected for 1850–1949.

**L90-91** I don't understand what this "new climatology" is. Please describe in more depth what this sentence means.

**L91** "resprective" -> "respective"

**L93** "that" -> "which"

**L97-97** The constants $C$ and $\gamma_T$ need units specified.

**L99–100** The decision to depart from observations for the Amundsen Sea due to a qualitatively undesireable model behavior merits some additional consideration. Why is lowereing the temperature necessary? Does this imply that the ocean model is doing something wrong, or that the ice sheet model is doing something wrong? What are the ramifications of this, and how much faith should we have in $C$ and $\gamma_T$ under this alteration?

**L107** "We here exemplify ..." is a weird sentence that can probably be removed.

**L114** The two configurations are similar in ice thickness, volume, and speed, but how about in trend? This is hinted at, but should probably be specified more explicitly.

**L115–119** This whole paragraph is a bit unclear. What does "these" refer to in the third sentence? What is "drift in the initial configuration"? What is an "increase in ocean forcing"?

**L124** An increase in mass balance would typically imply a lower mass loss rate, but I don't think that's what is meant here. Try to be more explicit about the signs of the figures being reported.

**L125–126** I don't understand this sentence. How is "more realistic" defined?

**Table 2** I think that there is a typo here: the text says ISMIP6, but should read LARMIP-2, unless I am gravely misunderstanding something.
**L209** I would like to see a more specific description of how the step-forcing experiments mentioned here were performed and analyzed; as it stands, the reader is left to extrapolate from Fig. 6 how these numbers were derived.

**Sec. 4.3** This paragraph seems somewhat underdeveloped, given that the role of historic trajectory is one of the key points of the paper (it's in the title!). Is there a strong trend baked into each simulation? Is there a way to analyze whether the historic model hits a tipping point that the pseudo-steady model doesn't? There must be a reason behind why this difference exists.

---

## Author Comment (AC1) · 11 May 2020

Please find our response to all reviewer comments in the attached PDF. The changes in the manuscript (generated by latexdiff) can be found at the end of the file.

Please also note the supplement to this comment:
https://www.the-cryosphere-discuss.net/tc-2019-330/tc-2019-330-AC1-supplement.pdf
* * *

---

## Author Response (AR1)

**Response to Reviewer Comments**
**Date: 2020-05-11**
**By R. Reese, A. Levermann, T. Albrecht, H. Seroussi and R. Winkelmann**

Journal: TC

Title: The role of history and strength of the oceanic forcing in sea-level projections from Antarctica with the Parallel Ice Sheet Model

Author(s): R.Reese, A.Levermann, T.Albrecht, H.Seroussi and R.Winkelmann

MS No.: tc-2019-330

MS Type: Research article

First of all, we would like to thank the editor Douglas Brinkerhoff and the two anonymous reviewers for their helpful and excellent comments. We provide detailed replies to all comments below, with our responses indicated in blue. Line numbers refer to the revised version of the manuscript.

**Anonymous referee 1**

**General comments:**

This paper describes the sensitivity of ice-sheet projections to the initialization method (simulating the 1850-2014 historical period vs starting in 2015, after the same long spin-up in both cases). It also describes the sensitivity to ocean warming, and compares the effect of parameterised melting (through PICO in ISMIP6 simulations) to melt perturbations imposed in LARMIP2. These sensitivities are expressed as Antarctic contributions to sea level rise.

The paper is well written, although a few clarifications are required in the Methods section. While I find the results useful for the ice-sheet/climate community, I have a few major comments that should probably be addressed before publishing this paper:

We would like to thank the anonymous reviewer for his/her effort to review our manuscript and greatly appreciate his/her comments for improving our study.

1. An important conclusion of this paper appears to be that ice-sheet models should simulate the entire historical period to make meaningful projections. However, this would not be true is the initial state was selected in a way to get present-day thinning rates (e.g. including an observational ice thinning rate in the score used to pick the initial state in the long-term ensemble). I don't expect the authors to change this in their methodology, but this should be discussed.

Thanks for making the suggestion to add thinning rates to the scoring system - we want to consider this in the future as also discussed in the specific comments. Since in the initial ensemble we run the members for several thousand years with constant climate conditions towards equilibrium, we think that ideally this would be combined with a historic simulation that yields correct thinning rates for present-day, i.e., making 'hindcasting' experiments.

We believe that getting thinning rates correctly in the initial state would substantially improve projections. However, due to the non-linearity of the equation system, this might not be fully sufficient. For example, appropriate estimates of the three-dimensional temperature field would be required.

We added this to the discussion in Section 4.3. For a detailed discussion of the effects of the historic simulation, please see the reply to the last comment of reviewer 2.

2. A strong ad hoc temperature correction is applied to the Amundsen Sea, with deep-ocean temperatures set to $-1.25°$C, i.e. relatively cold conditions. Then, it is claimed that the PICO tuning parameters cannot be changed because they were tuned to match observational ice-shelf averaged melt rates (in Reese et al. 2018 with other ocean forcing dataset). These aspects need to be investigated in this paper. Melting (and ideally the ice-sheet response) without temperature correction should be described, and other tuning parameters should be considered. Further, it could be argued that the important observational target in terms of ice-shelf buttressing is not the ice-shelf average but the average in PICO's first box (near grounding line).

Many thanks for pointing us to this inconsistency. In line with Jourdain et al. (view) and Seroussi et al. (view), a central finding of this manuscript is that the tuning of basal melt rate parameterisations should be reconsidered for future projections of sea-level rise. Using a temperature correction in the tuning is one approach that we plan to explore in the future also for other basins than the Amundsen Sea. We changed the text correspondingly in lines 154ff and lines 285ff. We agree that tuning the parameterisation to obtain sub-shelf melt rates in PICO's first box would be a great addition for future work, as well as considering melt rate variability and temperature corrections (added in lines 288f).

**Table 1:** Amundsen Sea ocean temperature $T$, average basal melt rates $\dot{m}$ and melt rate sensitivity $\Delta\dot{m}$, calculated from one degree of ocean warming.

| Experiment | year | $T$ (°C) | $\dot{m}$ (m a$^{-1}$) | $\Delta\dot{m}$ (m a$^{-1}$ °C$^{-1}$) |
|---|---|---|---|---|
| Pseudo steady-state | 1850 or 2014 | $-1.25$ | 4.9 | 8.3 |
| After historic | 2014 | $-0.65$ | 9.9 | 8.6 |
| WOA 2018 pre-release | 1955-2018 | 0.85 | 23.1 | 9.2 |
| Schmidtko | 1975-2012 | 0.45 | 19.9 | 9.1 |

To estimate the effects of the temperature correction in the Amundsen Sea, we calculated the melt rate and melt rate sensitivity with and without the temperature correction (Table 1 here). The different inital ocean conditions in the Amundsen Sea yield different basin-wide average melt rates. However, as also shown in Figure 7 of the manuscript, the melt-rate sensitivities in PICO are (especially for warmer temperatures) mostly linear. Hence, basal melt rate changes are predominantly controlled by temperature changes and less by initial temperatures which means that our results are not much influenced by the temperature correction applied in the Amundsen Sea.

We show a constant-climate simulation for $5,000$ years with the 'warm' ocean conditions

[Figure]

**Figure 1:** Ice thickness change and grounding line movement for warm Amundsen Sea conditions. The simulation is simular to the pseudo-steady-state reference configuration, except that Amundsen Sea temperatures are at the warm, present-day values. Results are similar for all members of the initial ensemble.

for comparison (see Fig. 1 here) as well as projections for different PICO parameter (see Fig. 2 of this response). Using 'warm' ocean conditions in the Amundsen Sea in the inital ensemble, we find for all ensemble members, that the WAIS starts to collapse. Given that the current imbalance of the Amdunsen Sea region is likely caused by the ocean, we think that for the initialisation procedure employed in this study, a reduction of ocean conditions to 'cold' is a feasible assumption: this correction does not mean that PICO produces overly high melt rates for the Amundsen Sea region (average melt rates are in line with observations, see Reese et al. (2018) and Table 1 of this response) or that PISM cannot handle observed melt rates. The response of PISM to melt rates close to present-day in the Amundsen sea is a collapse of that region over several thousand years of constant climate forcing (see Figure 1 of this response). With this region is out of balance today, creating

a steady state with present-day ocean conditions would require to tune parameters to be overly stable. This is also in line with other model studies (e.g., Favier et al., 2014). This is discussed in detail in the reply to the specific comment on lines 99-100.

We test for different PICO parameters by re-running projections forced with NorESM1-M (Fig. 2) with increased parameter values. We find that, depending on the parameters, mass loss increases substantially. This is in comparison to control simulations which show a large drift since the initial setup is not close to equilibrium with the new parameters. For future sea-level projections, a thorough re-assessment of the parameters would be necessary.

Please see also the detailed reply to the specific comment below and to the corresponding comment by Reviewer 2.

3. Melt rates obtained through the ISMIP6 framework are compared to the melt rates imposed in LARMIP2 (based on previous plume-model studies). It would be interesting to describe how these two types of melt rates compared to other observational or modelling studies (see specific comment below).

Thanks for mentioning this. Please see the repsonse to the specific comment on Section 4.2 below.

**Specific comments:**

L. 43-45: These are global feedbacks related to freshwater injection into the ocean. There are also important local thickness-melting feedbacks described by Timmermann and Goeller (2017) and Donat-Magnin et al. (2017).

Many thanks for pointing us to these two relevant studies.

L. 51: 'haven' -> 'have'. Done.

Section 2.1: I don't necessarily ask the authors to do it, but it would be interesting to see the score of the historical simulation in 2015 compared to the ensemble of initial states and in particular the one that was selected against present-day observations of ice geometry and speed.

Please see the new Supplementary Figure 1 in which we added the historic simulation to the scoring. It scores slightly worse than the selected configuration. As you mention below, using present-day thinning rates would be a helpful measure to test the historic simulation, which is not reflected in the current scores. Please see also the reply to the respecive comment by Reviewer 2.

L. 81: About 'scoring against present-day observations of ice geometry': including the ice thinning rate in the score would have helped get an initial state similar to observations and maybe to the historic state at 2015.

This is a good idea that we will add for future studies.

L. 89-90: What constant value is used for $1850 - 1949$?

We added an explanation in lines 130f. Note that the used, aggregated, yearly surface mass balance is very similar to the RACMO climatology as shown in the historic versus the ctrl*-simulation in Fig. S3.

L. 90-91: This sentence is unclear.

Please see the changed text (lines 134-140) and the response to the respective comment by Reviewer 2.

Section 2.2: Specify here what is done for SMB in LARMIP-2. Done.

L. 96-97: '$C = 106$' would be better than programming notation. And both $C$ and $\gamma_T$ should have units. The $\gamma_T$ value does not seem to be exactly the same as Reese et al. (2018), why has it been modified despite a careful tuning in that paper?

We changed the notation of $C$ and added units.

The value of $\gamma_T$ was part of the initial ensemble. We added this to the manuscript in lines 90ff as well as a Supplementary Figure showing the scores of the ensemble. The range of $\gamma_T$ in the ensemble has been based on Reese et al. (2018), where diagnostic experiments were done to tune the parameters. We here found that in transient simulations the higher value to yield a better initial configuration. However, as we discuss also in the response to your major comments, this value yields sensitivities lower than the ones used in LARMIP-2. A re-evaluation of the basal melt rate sensitivity and an alternative tuning approach might be needed. Note that the findings for the Amundsen Sea region (major comment 2) were consistent for all values of $\gamma_T$ in the initial ensemble.

L. 98-99: How was the data compilation based on World Ocean Atlas (WOA) 2018 and Schmidtko's dataset built? Was the latest update of WOA 2018 used? If not it is worth being mentioned because there were important updates near Antarctica.

See https://www.nodc.noaa.gov/OC5/woa18/woa18data.html

Thanks for pointing this out. We did use the pre-release of WOA2018 and added this correspondingly as well as a more detailed description of how the data from WOA2018-prerelease and (Schmidtko et al., 2014) were processed and combined. The exact procedure is described in the Bachelor Thesis by Lena Nicola (2019) - we are happy to send it to everyone interested.

L. 99-100: Changing the Amundsen Sea temperatures to $-1.25°$C is quite an issue. Does it mean that PICO produces overly high melt rates in the Amundsen Sea, or that PISM cannot handle melt rates close to observational estimates? In Levermann et al. (2020), a similar correction is done to $-0.37°$C with a claim that this possibly represents pre-industrial conditions. This needs to be discussed, and what's happening for uncorrected temperatures needs to be shown in this paper. It is a crucial point because (1) the Amundsen Sea is our

best present-day test for future warm conditions, and (2) the small sensitivity reported in this paper with the ISMIP6 framework could be due to this artificial cooling in the Amundsen Sea.

Thanks for bringing up these points which is very important. This is an ad-hoc correction to reduce Amundsen Sea temperatures to 'cold' conditions that we discuss in the following and in lines 154ff of the revised manuscript. As explained in out reply to the general comment 2, this does not mean that PICO produces overly high melt rates for the Amundsen Sea region.

Concerning (1): especially because the Amundsen Sea is our best present-day test for warm conditions and since observations show that the Amundsen Sea is out of balance at the moment, we argue that reducing ocean temperatures in the Amundsen sea is important due to the initialisation procedure of our experiments (see above). Furthermore, we think that observations of the Amundsen Sea melt rates would be very valuable for assessing PICO's melt rate sensitivity in a next step.

Concerning (2): note that in PICO, the changes in basal melt rates are mostly dependent on the changes in ocean temperatures and not so much on the initial ocean temperature, since the sensitivity of basal melt rates to ocean temperatures is mostly linear. This is true especially for larger ocean temperatures, see Figure 7 in the manuscript and Figure 6 in Reese et al. (2018). Please note that over the historic simulation, Amundsen Sea temperatures increase by 0.6°C, see Table 1 in this response, so that the projections based on the historic simulation start from warmer ocean conditions in the Amundsen Sea.

We agree with the reviewer that the correction of temperatures as well as the melt rate sensitivity should be further assessed in a new tuning approach for basal melt rate parameterisations for projections of Antartcica's future mass loss.

Please see also the reply to your major comment 2 and the respective comment by Reviewer 2.

L. 103: Is NorESM1-M used from 1850 onward? I don't understand what is the 'new climatology' and how it is used.

Similar to the atmosphere forcing, we apply NorESM1-M ocean forcing in the historic simulation. To account for changes in the ocean temperatures and salinity over the historic simulation, we initialize the projections based on the historic simulation from a new climatology. This climatology is obtained from the 1995-2014 average conditions in the historic simulation, to make sure that we do not start simulation, e.g. the control run, from exceptionally high or low values at the end of 2014 that arise from interannual varibility in the forcing (see Fig. S3).

Note that we decided to start the historic simulation from the same conditions as the pseudo-steady state which means that atmospheric and oceanic boundary conditions in 2015 differ between the two configurations. However, this allows to start from two states

that are very similar (see lines 103ff) and avoids running two ensembles with different boundary conditions, one with present-day conditions and one with scaled historic conditions. Since changes over the historic simulation are small, see Fig. S3 we expect this to have a minor effect.
We added an explanation in lines 161f.

L. 104: Not clear, what 'ISMIP6 ocean forcing' refers to? It has just been explained that PICO is used, 'initialized with an ocean data compilation from Locarnini et al. (2018) and Schmidtko et al. (2014)', with additional CMIP5 anomalies. Is there anything else to add?
We wanted to make clear that the projections are run with ocean forcing only. We added a new section (2.2) to describe the experiments in order to clarify this.

● Arriving at section 3, it is surprising to see several CMIP5 models while only NorESM1-M was mentioned in section 2. The use of several models (all starting from NorESM1-M's 1850-2014 period) should be described in section 2. The methods should be understandable without knowing the full ISMIP6 framework.
We added a new subsection (2.2) and a table S1 to describe the experiments in more detail.

L. 124: 'basal mass balance increased' or basal mass balance DEcreased? Same in Tab. 1, should SMB and BMB have the same sign for ice-sheet mass loss? Maybe just a matter of taste...
Done.

Section 3.1: it might be clearer to define the 'ctrl' experiment in section 2, and to explain how it is designed: which part of the historical forcing is kept constant until 2100?
We added a description in the new subsection 'experiments' of Section 2.

Tab.1 and Sect. 3.2: please briefly define 'asmb' and 'abmb' so that it can be understood without having to read Seroussi et al. (2019).
Explained in the new section 'experiments'.

Throughout Sect. 3: In view of the drifts in ctrl vs ctrl*, it is expected to see different mass loss for the two initialisations. All the plots and mass loss estimates could be calculated with respect to ctrl or ctrl*... Done.

L. 146: you could cite Edwards et al. (2019) here as it revisits several previous estimates. Done.

L.147: it is difficult to visualise the 50%. You could consider redo Fig. 2 with ctrl and ctrl* subtracted.
Mass loss in Figure 2 is shown with respect to ctrl or ctrl*, respectively. We hope that this clarifies the comment?

Section 3.4: The differences between the initial state used for PISM-PIK in Levermann et al. (2020) and the one used for LARMIP-2 PISM simulations should be summarised somewhere in this manuscript. Without this it is difficult to understand what to expect from this section. We added a detailed comparison of the PISM-PIK contributions for LARMIP-2 and IS-MIP6 to the Supplementary Information (new Table S2) and refer to it in line 212f.

Section 4.1: here only NorESM1-M results are shown because it 'shows highest mass losses of all ISMIP6 experiments'. This is true for the total Antarctic mass loss, but the results considering individual sectors (as in Fig. 4) may be higher in other models. Furthermore, LARMIP-2 is based on a CMIP5 19-model mean, so it seems more appropriate to use the ISMIP6 multi-model mean here rather than just NorESM1-M.
Thanks for raising this point. We added in Fig. 4 and 5 basal melt rates and mass loss in all regions for the other RCP8.5 simulations following the ISMIP6 protocol that we present in this manuscript. Since basal melt rate forcing and mass loss in the experiments driven by CCSM and MIROC are overall similar to NorESM1-M, we can generalize our findings to these models. We would expect the multi-model mean of these models to yield similar results.
One interesting feature that arises when comparing the CMIP5-forcings is that basal melt rate changes in East Antarctica seem similar in Figure 4 for NorESM1-M and MIROC but mass loss is higher for NorESM1-M as shown in Figure 5. This is because ocean forcing in the ISMIP6 simulations varies across the different ice shelves in East Antarctica, but is constant for LARMIP-2. While there is stronger ocean warming in Dronning Maud Land and Amery in the MIROC forcing, the ocean warms substantially more in the Totten region for NorESM1-M. Since the Totten region is more vulnerable to ocean warming, melt rate increases in that area cause larger overall mass losses.
We added a description in lines 254ff.

L.224-225: It is true that tuning $C$ and $\gamma_T$ would 'affect the comparison of sub-shelf melt rates to present-day observations'. But (1) the input temperatures have been modified compared to the initial tuning of PICO, and (2) it could also be argued that the important observational target is not the ice-shelf average but the average in PICO's box 1 (near grounding line). In view of Reese et al. (2018b), this is what matters the most for buttressing, isn't it?
Thanks for this point. We agree that the melting in PICO's first box should be added as a tuning parameter in future work. We changed and extended this discussion in lines 288f.

Section 4.2: The sensitivity used in LARMIP (7 to 16 m/yr/K) was estimated from plume models (Jenkins 1991 and Payne et al. 2007). It is discussed in the manuscript as if these were well-established values. These numbers may be acceptable, but there have been a few observational and more complex modelling studies since then, which have estimated sensitivities to ocean warming. It would be useful to mention whether these numbers can

still be considered as plausible (Jenkins et al., 2018; Naughten et al., 2018; Seroussi et al., 2017).

We estimate the sensitivity from the coupled experiments in Seroussi et al. (2017) for Thwaites glacier. The average sensitivity over the first twenty years is about $9.4\mathrm{m\,a^{-1}\,K^{-1}}$ which is closer to the estimates used in LARMIP-2.

Naughten et al. (2018) report that relevant ocean temperatures driven by ACCESS under the RCP85 scenario increase by 1.8°C in the Amundsen Sea. We estimate that melt rates for that scenario increase between about 6 to about $25\mathrm{m\,a^{-1}}$, which yields an rough estimate of 3.3 to $13.9\mathrm{m\,a^{-1}\,K^{-1}}$ (Fig. 7 of the paper). Especially for Pine Island and Thwaites glaciers ice shelves, melt rates increases are around $20\mathrm{m\,a^{-1}}$ which corresponds to $11\mathrm{m\,a^{-1}\,K^{-1}}$ which would be in line with the estimates used in LARMIP-2 and of (Seroussi et al., 2017). Similarly, Jenkins et al. (2018) show an estimate of how aggregate melt rates change quadratically based on observations for Dotson Ice Shelf (see Fig. 4c of the paper). If we assume an ice shelf area of $5,803\mathrm{km^2}$ (Rignot et al., 2013), this yields a sensitivity of about 3.8 to $12\mathrm{m\,a^{-1}\,K^{-1}}$ depending on initial temperatures. For the Filchner-Ronne Ice Shelf, Hellmer et al. (2012) observe an increase of basal melt rates from 0.2 to $4\mathrm{m\,a^{-1}}$ for a warming of 2K which would imply a sensitivity of $1.9\mathrm{m\,a^{-1}\,K^{-1}}$. A more thorough analysis of these studies is important for re-assessing parameters of PICO in future studies.

We tested the projection based on the NorESM1-M ocean forcing using different parameters for PICO that yield higher melt rate sensitivities. Figure 2 in this response shows that mass loss (relative to control simulations) increases substantially.

We added a dicussion of the sensitivies that we derived from (Seroussi et al., 2017) in lines 291ff.

Besides, it is possible that the ocean sensitivity depends on the ice shelf under consideration (for example because ocean heat entering a very large ice-shelf cavity will be entirely consumed, while only a part of the available heat is consumed in smaller cavities, which may be captured by PICO?). Please also discuss these aspects.

Done.

L.225-227: a similar correction has actually been applied to the Amundsen Sea in this paper.

Thanks for pointing this out - we changed the text accodingly.

[Figure]

**Figure 2:** ISMIP6 projections for different PICO parameters. Simulations are forced with NorESM1-M ocean conditions for 2015 to 2100 similar to Figure 1b in the paper. Results are shown with respect to the control simulations. Note that control simulations show large drifts due to the parameter changes in PICO.

**Anonymous referee 2**

**Summary**

In The role of history and strength of the oceanic forcing in sea-level projection from Antarctica with the Parallel Ice Sheet Model, Reese et al. present the results of applying a suite of numerical experiments associated with both the ISMIP6 and LARMIP-2 model intercomparisons to the parallel ice sheet model. The key findings of this paper are that 1) the inclusion of a century-scale climate history leads to a significantly different mass loss trajectory for all of the experiments included in ISMIP6 (in particular, a climate history leads to greater mass loss), and 2) that the mechanism for parameterizing sub-shelf melting matters alot, and that choosing between two different methods leads to an order of magnitude difference in century scale sea-level predictions.

We would like to thank the anonymous reviewer for his/her effort to review our manuscript and greatly appreciate his/her comments for improving our study.

**General comments**

Overall, I find the paper to be well-written and very interesting. The scientific quality is high, and the conclusions presented will be useful for those trying to hone in on areas of remaining uncertainty when prognosticating with regard to Antarctica. One overarching comment is that the experiments that are applied should be described more fully: because this paper deals with numerical experiments drawn from two other works (the main ISMIP6 forcing paper and

the LARMIP-2 paper, one of which is still in review), it would be helpful to briefly state the assumptions and differences between them (to aid in intercomparing the intercomparisons, as it were). My remaining comments are on a line-by-line basis, and may be found below.

We added a new subsection to describe the experiments and their differences in more detail, a Table S1 that contains all experiments (and the MIP's they belong to) and added more detail in the sections describing the forcing in the experiments, see, e.g., Sect. 2.2., 2.3 and 2.4. We hope that this makes it easier for the reader to understand our findings.

**Specific comments**

L80 A description of the mechanism for creating the ensemble, as well as the scoring method, would be appropriate here. Additionally, a discussion of the degree to which the optimal ensemble member actually matched observations would help in determining how seriously the reader should take the predictions included in this work.

We added here a detailed description of how the ensemble was created and how the initial configuration was selected in lines 87ff. The ensemble presented in the manuscript was based on a number of pre-tests of parameter influence and ranges. The ranges of PICO parameters were based on the diagnostic tuning in Reese et al. (2018). Testing for the full uncertainty related to model parameters in sea-level projections would require running the experiments on a broader ensemble.

The newly added Supplementary Figure S1 shows the scores for the ensemble. We added the state in 2015 after the historic simulation to the figure. The historic simulation score only slightly worse than the previously best run that was selected to be the initial configuration. We want to underline that given this slight difference, the final ranking depends of course on the specific choice of indicators. If, for example, present-day thinning rates would be included as suggested by Reviewer 1, we would expect the historic simulation to improve.

We added the root-mean-square deviation in ice-stream velocitiy and ice thickness as well as deviation in grounding line positions for both initial configurations to the manuscript in lines 104ff.

L86 Add "forcing" after "historic". Done.

L89 Please add a citation for NorESM1-M. Done.

L89-90 Please describe how the climate constants were selected for 1850-1949.

We added an explanation in lines 130ff.

L90-91 I don't understand what this 'new climatology' is. Please describe in more depth what this sentence means.

We added a detailed description in lines 137ff.
Note that the time period of the climatology $(1995 - 2014)$ is in line with time period used for the preprocessing of data in ISMIP6.

L91   "resprective" -> "respective" Done.

L93   "that" -> "which" Done.

L97-97   The constants $C$ and $\gamma_T$ need units specified. Done.

L99-100   The decision to depart from observations for the Amundsen Sea due to a qualitatively undesireable model behavior merits some additional consideration. Why is lowereing the temperature necessary? Does this imply that the ocean model is doing something wrong, or that the ice sheet model is doing something wrong? What are the ramifications of this, and how much faith should we have in $C$ and $\gamma_T$ under this alteration?
Thanks for bringing up this point. Please refer to the reply to the respective major and specific comments by Reviewer 1.

L107   "We here exemplify ..." is a weird sentence that can probably be removed. Done.

L114   The two configurations are similar in ice thickness, volume, and speed, but how about in trend? This is hinted at, but should probably be specified more explicitly.
Thanks for making this point, we added it in lines 172f.

L115-119   This whole paragraph is a bit unclear. What does "these" refer to in the third sentence? What is "drift in the initial configuration"? What is an "increase in ocean forcing"?
We rewrote the whole paragraph to make it better understandable.

L124   An increase in mass balance would typically imply a lower mass loss rate, but I don't think that's what is meant here. Try to be more explicit about the signs of the figures being reported.
We switched the signs so that basal mass balance is negative, i.e., a decrease means now higher mass loss, see line 179, Table 1 etc.

L125-126   I don't understand this sentence. How is "more realistic" defined?
We meant that present-day Antarctica is currently losing mass, hence the 'historic' initial configuration that has a tendency to lose mass is closer to observations (compared along that dimension) than the pseudo-steady state simulation that has a tendency to gains mass. We changed the text accordingly in line 182.

Table 2   I think that there is a typo here: the text says ISMIP6, but should read LARMIP-2, unless I am gravely misunderstanding something. Done.

L209 I would like to see a more specific description of how the step-forcing experiments mentioned here were performed and analyzed; as it stands, the reader is left to extrapolate from Fig. 6 how these numbers were derived.

We added a description of the step forcing experiments in lines 270f.

Sec. 4.3 This paragraph seems somewhat underdeveloped, given that the role of historic trajectory is one of the key points of the paper (it's in the title!). Is there a strong trend baked into each simulation? Is there a way to analyze whether the historic model hits a tipping point that the pseudo-steady model doesn't? There must be a reason behind why this difference exists.

We extended this section to discuss these hypotheses, added a new figure to the Supplementary Information showing the rates of sea-level contribution for the ISMIP6-simulations (Figure S6) and estimate the LARMIP-2 sea-level probability distribution for the pseudo-steady state configuration. The effect of the historic simualtion is less pronounced in the LARMIP-2 experiments, probably because the forcing is stronger, thereby reducing effects of internal dynamics. If a tipping point has been crossed in the simulation could be estimated by performing equilibrium and hysteresis experiments, which would be very interesting for a next study.

**References**

Favier, L., Durand, G., Cornford, S. L., Gudmundsson, G. H., Gagliardini, O., Gillet-Chaulet, F., Zwinger, T., Payne, A. J., and Le Brocq, a. M. (2014). Retreat of Pine Island Glacier controlled by marine ice-sheet instability. *Nature Climate Change*, 5(2):117–121.

[revised manuscript text omitted]

**Figure S.1.** Comparison of PISM ensemble members with present-day geometry and velocities (Fretwell et al., 2013; Rignot et al., 2011) after (upper row) 5,000 and (lower row) 12,000 years of model simulation. Scores are obtained as a product of normalized root mean square deviations from present-day ice thickness and ice speed, deviations in grounded and floating areas and grounding line positions, in line with the approaches presented in (Pollard et al., 2016; Albrecht et al., 2020). A focus is layed on the Amundsen Sea, Filchner-Ronne and Ross ice shelves by testing for those regions in particular. The individual scores are normalized to their median value with smaller scores indicating better fit with observations. The ensemble was done for PICO's heat exchange coefficient (left panels), PICO's overturning coefficient (middle panels) and the minimum till friction angle of the parameterized basal till properties (right panels). After 5,000 years, the best 5 simulations were continued and re-scored after 12,000 years to select the best ensemble member, shown in blue here with the state after the historic simulation shown in light blue.

[Figure]

**Figure S.2.**  Modeled ice thickness as in **(a)** present-day pseudo-equilibrium configuration, and **(b)** changes after the historic run. Simulated ice speed in **(c)** pseudo-equilibrium and **(d)** changes after the historic run. Black contours indicate the initial (a,c) and final (b,d) grounding line location.

[Figure]

**Figure S.3.** ISMIP6 experiments with (solid lines) and without (dashed lines) historic initialisation. Shown is the evolution of the (a) volume above flotation, (b) surface mass balance, (c) basal mass balance and (d) calving flux at the ice front relative to the starting condition. Experiments are forced with changes in ocean temperature and salinity and surface mass balance and temperatures from the ISMIP6 protocol experiments no. 1 and 3.

[Figure]

**Figure S.4.** Changes in ice thickness **(a)** with and **(b)** without the historic run between 2100 and 2015. The corresponding changes in ice speed **(c)** with and **(d)** without the historic run for experiment no. 1 from ISMIP6 (NorESM1-M, RCP8.5).

[Figure]

**Figure S.5.** Sensitivity of sub-shelf melt rates of Thwaites glacier in the coupled simulation from Seroussi et al. (2017). The sensitivity is estimated from the shelf-wide average melt rate in two coupled simulations that differ by initial and boundary ocean temperatures of $0.5\,^{\circ}$C. The sensitivity might be biased by differently evolving ice-shelf cavities over time.

**Table S.1.** List of all experiments, with INIT being based on the historic simulation starting from INIT*.

| MIP | INIT | INIT* |
|---|---|---|
| | | historic ctrl |
| | | historic |
| all | ctrl | ctrl* |
| initMIP | asmb | asmb* |
| initMIP | abmb | abmb* |
| ISMIP6 | NorESM RCP85 | NorESM RCP85* |
| ISMIP6 | MIROC RCP85 | MIROC RCP85* |
| ISMIP6 | NorESM RCP26 | NorESM RCP26* |
| ISMIP6 | CCSM4 RCP85 | CCSM4 RCP85* |
| LARMIP-2 | AP $4\,\mathrm{m\,a^{-1}}$ | AP $4\,\mathrm{m\,a^{-1}}$ |
| LARMIP-2 | EAIS $4\,\mathrm{m\,a^{-1}}$ | EAIS $4\,\mathrm{m\,a^{-1}}$ |
| LARMIP-2 | RS $4\,\mathrm{m\,a^{-1}}$ | RS $4\,\mathrm{m\,a^{-1}}$ |
| LARMIP-2 | AS $4\,\mathrm{m\,a^{-1}}$ | AS $4\,\mathrm{m\,a^{-1}}$ |
| LARMIP-2 | WS $4\,\mathrm{m\,a^{-1}}$ | WS $4\,\mathrm{m\,a^{-1}}$ |
| LARMIP-2 | AP $8\,\mathrm{m\,a^{-1}}$ | AP $8\,\mathrm{m\,a^{-1}}$ |
| LARMIP-2 | EAIS $8\,\mathrm{m\,a^{-1}}$ | EAIS $8\,\mathrm{m\,a^{-1}}$ |
| LARMIP-2 | RS $8\,\mathrm{m\,a^{-1}}$ | RS $8\,\mathrm{m\,a^{-1}}$ |
| LARMIP-2 | AS $8\,\mathrm{m\,a^{-1}}$ | AS $8\,\mathrm{m\,a^{-1}}$ |
| LARMIP-2 | WS $8\,\mathrm{m\,a^{-1}}$ | WS $8\,\mathrm{m\,a^{-1}}$ |
| LARMIP-2 | AP $16\,\mathrm{m\,a^{-1}}$ | AP $16\,\mathrm{m\,a^{-1}}$ |
| LARMIP-2 | EAIS $16\,\mathrm{m\,a^{-1}}$ | EAIS $16\,\mathrm{m\,a^{-1}}$ |
| LARMIP-2 | RS $16\,\mathrm{m\,a^{-1}}$ | RS $16\,\mathrm{m\,a^{-1}}$ |
| LARMIP-2 | AS $16\,\mathrm{m\,a^{-1}}$ | AS $16\,\mathrm{m\,a^{-1}}$ |
| LARMIP-2 | WS $16\,\mathrm{m\,a^{-1}}$ | WS $16\,\mathrm{m\,a^{-1}}$ |

AP = Antarctic Peninsula, EAIS = East Antarctica, RS = Ross Sea, AS = Amundsen Sea, WS = Weddell Sea as specified in (Levermann et al., 2020).

**Table S.2.** Comparison of the PISM-PIK LARMIP-2 contribution and the PISM-PIK ISMIP6 contributions.

|  | ISMIP6 | LARMIP-2 |
|---|---|---|
| horizontal resolution | 8 | 4 |
| vertical resolution | 13-100m | 7-48m |
| initialisation | steady-state, historic | 600a constant climate |
| sub-grid friction at the GL | yes | yes |
| sub-grid melt at the GL | yes | no |
| basal melt rates | PICO | PICO |
| atmosphere | RACMOv2.3 | RACMOv2.3 |
| ocean | WOA18+SCH14 | SCH14 |
| Amundsen temperature | $-1.25$ | $-0.37$ |
| till friction angle | parameterized (ensemble) | optimized |
| eigencalving | $K = 1 \times 10^{16}$ ms | $K = 1 \times 10^{17}$ ms |
| thickness calving | threshold $< 50$m | threshold $< 200$m |
| prescribed maximum extent | Bedmap2 | none |
| sliding law | pseudo-plastic exponent $q = 0.75$ | plastic ($q = 0$) |

References: RACMOv2.3 (Van Wessem et al., 2018), WOA18 (Locarnini et al., 2018), SCH14 (Schmidtko et al., 2014).

[Figure]

**Figure S.6.** Rate of sea-level rise between 2015 and 2100. We compare rates of sea-level rise for simulations driven by GCM ocean forcing with the corresponding model specified in the legend. Time periods when sea-level rates are larger in the simulations based on the historic simulation are indicated in green and periods when the simulations starting from the pseudo-steady state induce stronger sea-level rise are indicated in red.